# Human T-Cell Lymphotropic Virus (HTLV 1/2) in Ecuador: Time for Action

**DOI:** 10.3390/v17030446

**Published:** 2025-03-20

**Authors:** Miguel Angel Garcia-Bereguiain, Solon Alberto Orlando, Melissa Joseth Carvajal Capa, Manuel Gonzalez

**Affiliations:** 1One Health Research Group, Universidad de Las Américas, Quito, Ecuador; 2Instituto Nacional de Salud Pública e Investigación, Guayaquil, Ecuador; 3Universidad Espíritu Santo, Guayaquil, Ecuador; 4Universidad Católica Santiago de Guayaquil, Guayaquil, Ecuador; 5Universidad Ecotec, Guayaquil, Ecuador

**Keywords:** HTLV 1, HTLV 2, Ecuador, surveillance

## Abstract

The human T-cell lymphotropic viruses of type 1 and 2 (HTLV 1/2) are retroviruses with estimations of 10 million people infected worldwide. HTLV 1/2 viruses are endemic in South America where Indigenous and Afro American populations are considered of high risk. Although several case reports of HTLV 1/2 associated pathologies and some prevalence studies have been reported in Ecuador, the country lacks a national surveillance and control program, and no screening of blood or organ donors is currently done. We discuss the problems associated to HTLV 1/2 in Ecuador and propose a strategy to improve a surveillance and control program.

The human T-cell lymphotropic viruses of type 1 and 2 (HTLV 1/2) are retroviruses with estimations of 10 million people infected worldwide [1,2,3]. The HTLV 1/2 prevalence varies for different geographical areas, with hotspots in several low- and middle-income countries across Africa, Asia, and America [1,2,4,5,6]. HTLV-1 infections are mainly asymptomatic, although 4–5% of infected individuals develop an adult T-cell leukemia/lymphoma with a poor prognosis, and 1–3% develop a disabling myelopathy known as HTLV-1-associated myelopathy-tropical spastic paraparesis (HAM-TSP), in addition to other clinical outcomes like dermatitis or uveitis [1,7,8,9]. In contrast, HTLV-2 infection has not been clearly linked with any pathology [1,3,10]. The modes of transmission of HTLV 1/2 are similar to those of other retroviruses like HIV. The most important routes of transmission are mother-to-child transmission by breast feeding, condomless sexual intercourse, blood transfusion, and organ transplantation [1,6,11,12].

In South America, HTLV infection is endemic, and it has been reported in all 13 countries of the region. The introduction of HTLV-1 to South America was probably associated with the slave trade from Africa, and it is linked to the Afro-American population. HTLV-2 is considered to be ancestral, and it is associated with the Indigenous people of the Americas [13]. There are several reports from countries in South America like Brazil, Peru, and Colombia that indicate a prevalence of HTLV-1 infection of up to 13.9% and an HTLV-2 infection rate of up to 57% in high-risk groups, including Afro-Americans and communities of Japanese origin for HTLV-1 [4,6,13,14,15,16,17] and Amerindian tribes for HTLV-2 [14]. The overall prevalence of HTLV 1/2 infection in the general population has been recently estimated to be below 0.2% in Colombia [18]. In these countries, HTLV screening is currently performed in blood donors [6,7,16].

On the other hand, HTLV 1/2 surveillance is neglected in other countries of the region like Ecuador. According to a recent technical report from the European Center for Disease Control and Prevention regarding the HTLV 1/2 global distribution, Ecuador is considered to be a country with non-reliable data for HTLV 1/2 epidemiology [1]. The national reference laboratory for retroviruses from “Instituto Nacional de Salud Pública e Investigación” has not been performing HTLV 1/2 diagnosis for years. Moreover, HTLV 1/2 screening is not performed in blood or organ donors in Ecuador. The Ecuadorian public health authorities have not implemented a surveillance and control program for HTLV 1/2 to date. However, there are two reports in Ecuador showing that HTLV is endemic in Indigenous communities and Afro-American people, with prevalence values of 2.8% in a study from 1994 and 3.5% in a study from 2019 [19,20]. Additionally, there are several clinical case reports of HAM-TSP in Ecuador [21,22,23]. We call attention to a recent publication describing an HAM-TSP case following a kidney transplant from an HTLV-1-positive donor in a hospital from Quito [12]. So far, the available scientific literature supports that HTLV-1/2 are endemic and neglected viruses in Ecuador. The overall prevalence, the prevalence in certain ethnic groups at risk, and the incidence of HTLV 1/2-associated pathologies like HAM-TSP are totally unknown in Ecuador. The lack of prevalence studies makes the Ecuadorian public health authorities unaware of the risk associated with this neglected tropical disease. This is especially worrisome considering that underserved Indigenous and Afro-American communities represent around 10% of the Ecuadorian population, and those ethnicities are considered to be risk populations for HTLV 1/2 [20].

Considering this scenario, we call for immediate action by the Ecuadorian public health authorities to develop a national surveillance and control program for HTLV 1/2. The milestones to achieve this goal include (1) implementing state-of-the-art diagnostic tools, either serological or PCR-based, in the national reference laboratory of retroviruses from the “Instituto Nacional de Salud Pública e Investigación” to assist with laboratory confirmation of suspicious clinical cases across the country; (2) implementing serological screening in blood banks and organ donors across the country to prevent HTLV 1/2 transmission through transfusions and transplants; and (3) implementing an active case-finding program of HTLV 1/2 cases for underserved Indigenous and Afro-American communities, with a special focus on rural and remote communities without access to medical care. It is important that active case-finding programs guarantee follow-up and health counseling for infected individuals and also community-oriented educational programs to prevent stigmatization. These approaches would be the pillars of a sustained and effective HTLV 1/2 surveillance and control program to aim for the reduction in the burden of this disease.

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
