# Peer review of "Human T-Cell Lymphotropic Virus (HTLV 1/2) in Ecuador: Time for Action"

_viruses, 2025, doi:10.3390/v17030446_

Round 1
Reviewer 1 Report
Comments and Suggestions for Authors
The authors make a convincing case for the development of a national surveillance and control plan for HTLV-1 and HTLV-2 in Ecuador. I invite the authors to address three concerns:
- I think reference 1 is outdated. In addition, the prevalence estimates presented in that reference may not be based on solid research methods. References 2 to 5 look more convincing.
- The prevalence estimates of HTLV-1 (13.9%) and HTLV-2 (57%) mentioned in line 28 seem very high. Maybe these come from small studies in special population groups? I suggest to also mention prevalence estimates in the general population and/or based on contemporary epidemiological and diagnostic methods.
- The final sentence reads like an overstatement: "Those approaches would be the pillars for a sustained and effective HTLV 1/2 surveillance and control program to aim for the reduction of the burden of this disease and its potential eradication." The jump from total neglect to potential eradication seems quite drastic.
Comments on the Quality of English Language
I suggest to revise the following sentence: "While HTLV-1 introduction was probably originated from the slave trade from Africa and it associated to Afro descendent population, HTLV-2 is considered ancestral and was brought in by immigrants through the Bering Strait and is widely distributed among Indigenous people of the Americas."
Author Response
Reviewer 1.
The authors make a convincing case for the development of a national surveillance and control plan for HTLV-1 and HTLV-2 in Ecuador. I invite the authors to address three concerns:
I think reference 1 is outdated. In addition, the prevalence estimates presented in that reference may not be based on solid research methods. References 2 to 5 look more convincing.
Thanks for this comment. We have now eliminated reference 1 and renumbered references accordingly.
The prevalence estimates of HTLV-1 (13.9%) and HTLV-2 (57%) mentioned in line 28 seem very high. Maybe these come from small studies in special population groups? I suggest to also mention prevalence estimates in the general population and/or based on contemporary epidemiological and diagnostic methods.
Thanks for this comment. We have rephrased this sentence to point out that those values are for the mentioned at risk ethnic groups. We have also added a recent reference for the overall prevalence in Colombia.
The final sentence reads like an overstatement: "Those approaches would be the pillars for a sustained and effective HTLV 1/2 surveillance and control program to aim for the reduction of the burden of this disease and its potential eradication." The jump from total neglect to potential eradication seems quite drastic.
Agree. We have eliminated "and its potential eradication".
Comments on the Quality of English Language: suggest to revise the following sentence: "While HTLV-1 introduction was probably originated from the slave trade from Africa and it associated to Afro descendent population, HTLV-2 is considered ancestral and was brought in by immigrants through the Bering Strait and is widely distributed among Indigenous people of the Americas."
Thanks for this comment. Not only this sentence but the whole manuscript has been revised to improve the quality of the English language.
Reviewer 2 Report
Comments and Suggestions for Authors
Though relatively few people infected with HTLV-1 develop complications, the virus has significant pathogenicity and public health measures are effective at reducing transmission. The issues raised by the authors in the context of Ecuador are therefore of considerable public health importance. The failure to test blood or organ donations in a country that is very likely to have high rates of infection are of particular concern.
The three actions called for by the authors are sensible. However, there could be concerns about active case finding of a potentially stigmatising infection in Indigenous and African American populations (Line 63). High prevalence rates have been reported for marginalized populations in many countries; however, active case finding can be problematic without extensive consultation and the support of those affected. This should be considered by the authors.
The language could be more precise and addressing a number of minor points would improve the manuscript.
Line 16. Could include other disease associations, such as infective dermatits, uveitis.
Line 17. HTLV-1 associated myelopathy- tropical spastic paraparesis (HAM-TSP)
Line 19. Change to 'Modes of transmission of HTLV-1 are similar to those of other retrovirus like HIV'.
Line 21. Change ‘unsafe sexual habits’ to eg., condomless sexual intercourse.
Line 23. ‘While HTLV-1 introduction was’ might be better understood as ‘The introduction of HTLV-1 to South America’
Line 24. ‘it associated to Afro descendent population’ could be better worded. Need to be consistent with terminology for those of African descent.
Line 28-30. Separate populations at risk of HTLV-1 and HTLV-2
Line 44. TSP, change to HAM-TSP
Line 48. ‘Risk factors associated to’. The meaning here is unclear. ‘Risk factors for transmission associated with’? Or ‘prevalence within ethnic groups’?
Comments on the Quality of English Language
Could be improved as noted above.
Author Response
Reviewer 2.
Though relatively few people infected with HTLV-1 develop complications, the virus has significant pathogenicity and public health measures are effective at reducing transmission. The issues raised by the authors in the context of Ecuador are therefore of considerable public health importance. The failure to test blood or organ donations in a country that is very likely to have high rates of infection are of particular concern.
Thanks for the positive feedback.
The three actions called for by the authors are sensible. However, there could be concerns about active case finding of a potentially stigmatising infection in Indigenous and African American populations (Line 63). High prevalence rates have been reported for marginalized populations in many countries; however, active case finding can be problematic without extensive consultation and the support of those affected. This should be considered by the authors.
Thanks for this comment. We have emphasized that active case finding should be accompanied by public health policies to support the infected individuals and to prevent stigmatization.
The language could be more precise and addressing a number of minor points would improve the manuscript.
Line 16. Could include other disease associations, such as infective dermatitis, uveitis.
Corrected.
Line 17. HTLV-1 associated myelopathy- tropical spastic paraparesis (HAM-TSP)
Corrected.
Line 19. Change to 'Modes of transmission of HTLV-1 are similar to those of other retrovirus like HIV'.
Corrected.
Line 21. Change ‘unsafe sexual habits’ to eg., condomless sexual intercourse.
Corrected.
Line 23. ‘While HTLV-1 introduction was’ might be better understood as ‘The introduction of HTLV-1 to South America’
Corrected.
Line 24. ‘it associated to Afro descendent population’ could be better worded. Need to be consistent with terminology for those of African descent.
Corrected.
Line 28-30. Separate populations at risk of HTLV-1 and HTLV-2
Corrected.
Line 44. TSP, change to HAM-TSP
Corrected.
Line 48. ‘Risk factors associated to’. The meaning here is unclear. ‘Risk factors for transmission associated with’? Or ‘prevalence within ethnic groups’?
Corrected.
Comments on the Quality of English Language
Could be improved as noted above.
Thanks for this comment. Not only this sentence but the whole manuscript has been revised to improve the quality of the English language.